# Community perceptions of yellow fever and its treatment practices in regions with reported outbreaks in Uganda: A qualitative study

Lena Huebl[1,2]*, Aloysious Nnyombi[3], Patricia Apoko[4], Denis Okello[5], Eddy Walakira[3], Ruth Kutalek[1]

**1** Unit Medical Anthropology and Global Health, Department of Social and Preventive Medicine, Center for Public Health, Medical University of Vienna, Vienna, Austria, **2** Department of Tropical Medicine, Bernhard Nocht Institute for Tropical Medicine & I Department of Medicine, University Medical Center Hamburg-Eppendorf, Hamburg, Germany, **3** Department of Social Work and Social Administration, Makerere University, Kampala, Uganda, **4** The Lutheran World Federation Uganda Program, Kampala, Uganda, **5** Socioeconomic Data Center, Kampala, Uganda

* lena.huebl@bnitm.de

## Abstract

### Background

Yellow fever (YF), a mosquito-borne viral hemorrhagic fever, is endemic to Uganda and has caused numerous outbreaks in recent years. This study explored local perceptions of YF outbreaks among vulnerable groups in Uganda to inform future public health campaigns.

### Methodology

A qualitative study examined community perceptions of YF and its treatment practices. Data were collected in six districts where YF outbreaks were reported in 2010 and 2016. A total of 76 individuals participated, comprising 43 semi-structured interviews, 10 expert interviews, and 4 focus group discussions, including vulnerable groups of older adults ≥ 65 years and pregnant women. Data were analyzed using grounded theory.

### Principal findings

Participants often recognized jaundice but did not distinguish YF from other causes of jaundice, such as newborn jaundice, severe malaria or hepatitis. Nevertheless, participants still considered YF a deadly disease. It was perceived to be transmitted through multiple pathways, including mosquito bites, airborne transmission, close contact with sick individuals, sexual intercourse, vertical transmission during pregnancy, poor hygiene, and certain foods. Treatments ranged from herbal remedies to visiting health centers. Several YF survivors shared first-hand experience, often relying on traditional medicine due to limited access to health facilities, diagnostic

**Data availability statement:** Relevant data are included in the manuscript as citation excerpts. Full interview transcripts cannot be shared publicly because of ethical reasons (public availability would compromise study participants' privacy). Data are available from the Ethics Committee of the Medical University of Vienna (Borschkegasse 8b/E06, 1090 Vienna, Austria, +43 140400 21470, ethik-kom@meduniwien.ac.at) or from the Center for Public Health at the Medical University of Vienna (Kinderspitalgasse 15/ 1. Stock, 1090 Vienna, Austria, +43140160-34881, https://publichealth.meduniwien.ac.at/ueber-uns/kontakt/) or from the Bernhard Nocht Institute for Tropical Medicine, Bernhard-Nocht-Straße 74, 20359 Hamburg, Germany, +49 40 4285380-0, bni@bnitm.de) for researchers who meet the criteria for access to confidential data.

**Funding:** The author(s) received no specific funding for this work.

**Competing interests:** The authors have declared that no competing interests exist.

options, and no specific treatment for YF. In remote areas, participants often did not know the cause of the outbreak, as awareness campaigns focused on symptoms, prevention, and mass vaccination.

## Conclusions/significance

If YF is not seen as a distinct disease entity, implementing diagnostic and preventive measures may be impeded. Moreover, failure to diagnose YF in clinical settings can hamper timely outbreak response. We recommend strengthening health literacy through health education and public participation in vulnerable communities with programs tailored to local needs, given that other infectious diseases are prevalent in the region. Furthermore, we propose that access to diagnostic testing for YF may be enhanced.

## Author summary

Yellow fever is a life-threatening disease transmitted by mosquitoes, with frequent outbreaks in Uganda despite a highly effective vaccine. This study addresses disease perceptions and treatment practices in communities affected by yellow fever outbreaks. Through qualitative analysis of interviews and group discussions with vulnerable groups (pregnant women and older adults) and medical experts we explored these perceptions aiming to improve future public health campaigns. Our findings showed that for most participants, yellow fever was a novel and deadly disease. Participants understood who is most susceptible to yellow fever, but their knowledge on the underlying cause, transmission, and prevention often overlapped with other diseases. Survivors often relied on herbal medicine due to limited access to healthcare and diagnostics. Furthermore, we found that there was an unequal access to health information, particularly in remote areas where people were often unaware of the cause of the outbreak, as awareness campaigns focused mainly on the symptoms and prevention. This study contributes to broader efforts in global health by emphasizing the importance of engaging communities in disease control programs, through health education and public participation, in addition to enhancing yellow fever diagnosis in the clinical setting.

## Introduction

Yellow fever (YF) is a viral hemorrhagic fever transmitted by *Aedes* or *Haemagogus* species mosquitoes and is endemic to tropical regions of South America and Africa [1]. The disease can be prevented with a live attenuated vaccine [2]. Despite the availability of safe and effective vaccines since the 1930s, outbreaks have persisted, and 40 countries are considered endemic and at high risk for YF outbreaks [3]. It is estimated that 109,000 severe cases and 51,000 deaths occur annually, with the

majority of the global disease burden (92.2%) occurring in Africa [4]. The case number is most likely underestimated, given that most YF cases are mild and, therefore, undetected due to nonspecific symptoms and limited surveillance or laboratory capacity in many at-risk regions [5]. It is estimated that in Africa, only 13% of YF infections are severe, with an overall mortality rate of 46% [6]. Thus, small YF outbreaks may go unnoticed due to the inability of surveillance programs to identify asymptomatic or mild YF cases [7]. The disease manifests in three phases. The initial infection phase is characterized by flu-like symptoms and viremia. The disease then enters remission, with patients improving and seroconversion. The final intoxication phase affects 15–25% of symptomatic patients. It is characterized by deterioration in clinical conditions; hemorrhage; multiorgan dysfunction with signs of acute liver failure, such as jaundice and hepatic encephalopathy; renal failure; shock; and death [8]. At present, no specific antiviral treatment is available for YF [1]. Since 2016, there has been a notable increase in YF outbreaks, both in nonendemic and endemic areas with historically low YF virus activity [9,10] and low vaccination coverage [11]. YF outbreaks are on the rise due to numerous factors, including climate change, migration, displacement, and inadequate routine immunization caused by overburdened healthcare systems dealing with competing public health priorities, as well as unstable governments and regional conflicts [5]. Furthermore, the resurgence of YF cases in East Africa is likely attributed to increased sylvatic transmission, which usually occurs after prolonged periods of inactivity and inadequate immunization coverage [12]. Additionally, deforestation may facilitate YF transmission by reducing biodiversity and increasing mosquito movement between forest fragments, thereby increasing infection risk [13].

Uganda, a low-income and impoverished country [14], faces numerous challenges, including frequent epidemics of viral hemorrhagic fevers (VHFs), particularly Ebola virus disease (EVD) [15], and more recently, Mpox [16]. These outbreaks strain an already overburdened healthcare system. In 2018, Uganda reported the highest number of disease outbreaks in the African region of the World Health Organization (WHO), with YF being the third most common [17]. In addition, Uganda currently hosts 1.6 million refugees, representing one of the largest refugee populations globally, with the majority residing in economically underdeveloped regions in northern Uganda [18].

In 2010, Uganda reported its most significant outbreak of YF to date. A total of 13 cases were laboratory-confirmed, while 250 cases were compatible with the clinical case definition, and 55 deaths occurred [19]. In response to the 2010 YF outbreak, an emergency mass vaccination was carried out in January 2011, targeting over 905,000 people in the northern districts of Abim, Agogo, Kitgum, Lamwo, and Pader [20]. In 2016, three smaller YF outbreaks were detected in the districts of Masaka and Kalangala in central Uganda and Rukungiri in southwestern Uganda [21]. A total of 65 suspected cases of YF have been reported, with seven confirmed cases and three deaths [22]. As part of reactive emergency mass vaccination campaigns, 627,706 individuals were vaccinated against YF in the aforementioned three districts [22]. Since then, numerous YF outbreaks have been documented in various districts of Uganda, including Masaka and Koboko in 2019; Buliisa, Maracha, and Moyo in 2020; and Wakiso in 2022 [23]. In each instance, a mass vaccination campaign was initiated [23]. At that time, the estimated overall population immunity was relatively low (4.2%) in Uganda because the vaccine was not incorporated into the routine immunization program [23]. In response to the re-emergence of YF outbreaks worldwide, a comprehensive global strategy to eliminate yellow fever epidemics by 2026 (the EYE strategy) was developed in 2017 by a coalition of partners, encompassing the World Health Organization's Strategic Advisory Group of Experts on Immunization (SAGE), Gavi the Vaccine Alliance, and the United Nations Children's Fund (UNICEF) [3]. The three main goals of the EYE strategy are to protect at-risk populations, prevent the international spread of the disease, and contain outbreaks rapidly [3]. For high-risk nations such as Uganda, the EYE strategy recommends a three-pronged vaccination strategy comprising routine immunization, phased mass vaccination campaigns and catch-up campaigns to maintain high population-level immunity [3]. In October 2022, the YF vaccine was introduced into the routine immunization program for children in Uganda [24]. This was succeeded with two phased mass vaccination campaigns, planned between 2023 and 2024, aimed at ensuring vaccine coverage for all age groups [25,26].

Despite the surge in YF, few studies have been conducted on the perception of YF disease in high-risk regions of sub-Saharan Africa [27–29], which represents a significant gap in the literature. This study aims to gain insight into the

perceptions of YF outbreaks among vulnerable groups in Uganda to inform future public health campaigns. This article is part of a broader investigation into YF and its perception in areas of reported outbreaks in Uganda. Thus, the research described in this paper explores perceptions of YF symptoms, perceived outbreak causes, transmission modes, diagnostic pathways, and treatment-seeking behaviors among people affected by YF outbreaks. To the author's knowledge, no studies have been conducted on the perception of YF and its treatment practices in Uganda.

## Methods

### Ethics statement

This study adhered to the principles outlined in the Declaration of Helsinki and followed Good Clinical Practice guidelines. Ethical approval was granted from the Uganda National Council for Science and Technology (SS4362), the Mildmay Uganda Research Ethics Committee (#REC REF 0504–2017), and the ethics committee of the Medical University of Vienna, Austria (EK 1284/2017). Permission to conduct the study was obtained from the District Health Officers of the following districts: Masaka, Kalangala, Rukungiri, Lamwo, Pader and Kitgum. Participation was voluntary. All study participants provided written, informed consent or witnessed oral consent and gave a thumbprint if the participants were not literate. All the participants permitted us to audio-record interviews and group discussions.

### Study design

We conducted a qualitative study using semi-structured interviews, expert interviews, and focus group discussions (FGDs). Field work was carried out between August and December 2017. This study used a grounded theory approach inspired by Strauss & Corbin to generate and understand emerging themes and patterns about local perceptions of YF [30]. The aim of this study was to explore community perceptions of YF and its treatment practices in regions affected by YF outbreaks in Uganda in 2010 and 2016. This study was part of a broader research project on YF in Uganda, investigating the largest YF outbreak to date in Uganda as well as three subsequent smaller outbreaks. We previously described the methods in detail and reported the main findings on perceptions of YF emergency mass vaccinations in *PLoS Neglected Tropical Diseases* [31].

### Study setting

The study encompassed two sites. Study site 1 comprised the districts of Kitgum, Lamwo and Pader in northern Uganda, where the largest YF outbreak in Uganda occurred in 2010. At the time of the outbreak, the population had recently returned home from internally displaced persons camps, having been displaced by a two-decade-long civil war [19]. In addition, this region had previously experienced large epidemics of EVD [32] and hepatitis E [33] during and after the Civil War.

Study site 2 included the districts Masaka and Kalangala in central Uganda and Rukungiri in southwestern Uganda, where three smaller YF outbreaks occurred in 2016 [21]. The second study site was not affected by that civil war and was characterized by a higher literacy rate [34] and better health indicators [35]. All districts mentioned above had large-scale YF awareness campaigns as part of emergency mass vaccination in response to YF outbreaks in 2010 and 2016, respectively. Furthermore, at both study sites, all persons over six months of age, including pregnant women, were eligible for free YF mass vaccination [20,22].

### Participant selection

We included vulnerable groups, comprising of people over 65 years of age and pregnant women who experienced the YF outbreak in 2010 or 2016 in their respective districts. This population is typically excluded from YF vaccination except during mass immunization to contain an outbreak. The participants had to be older than 18 years and were purposively

selected from the six districts previously identified as having been affected by YF outbreaks. In addition, participants had to have been vaccinated during the aforementioned emergency immunization campaigns and, therefore, were likely exposed to YF awareness efforts, which were part of the vaccination mobilization strategy. This study focused on perceptions of YF disease and was part of a larger research project on YF in Uganda that also investigated perceptions of emergency mass vaccination. Older men and women were included at both study sites to explore perceptions among vulnerable groups. However, pregnant women were included only at study site 2, where data were collected one year after the YF mass immunization campaign, allowing verification of gestational age at the time of YF vaccination using both the vaccination certificate and the child's age. Throughout the manuscript, we use the term "pregnant women" to refer to women who received the YF vaccination during their pregnancy. Expert interviews were conducted with health officers and medical personnel from districts at both study sites.

## Data collection

The District Health Officer (DHO) of each district had to grant permission to collect data in their district. Four field assistants familiar with the study sites and native speakers of the respective languages (Luganda, Rukiga and Acholi) assisted with selecting participants, data collection and translation. In addition, local leaders and village health teams helped recruit participants. We conducted semi-structured interviews and FGDs with pregnant women, older men, and older women. The interviews were performed in or around the participant's home. FGDs were conducted at health centers and community centers. Semi-structured interviews and FGDs were conducted face-to-face in their preferred native language, and participants were compensated for their time. Expert interviews were scheduled beforehand, conducted in English at their office and received no compensation. This study is part of a larger project, and the interview guides for semi-structured interviews, FGDs, and expert interviews are published in the supplementary materials of Huebl et al. (2024) *PLoS Neglected Tropical Diseases* [31].

We used the COREQ guidelines to enhance transparency and rigor of the study's reporting [36]. Two experts in the field reviewed our interview guides and questions were revised according to their feedback on clarity and relevance. After pilot testing in the field the interview guides were adapted again after the first three interviews. We used methodological triangulation through semi-structured interviews, FGDs, and expert interviews to ensure data quality and credibility [37]. In addition, we used data triangulation by collecting data from different populations, affected regions, and YF outbreaks. The data were analyzed independently, and the findings from semi-structured interviews, expert interviews, and FGDs demonstrated consistency across data sources and methods. However, it has to be pointed out that in northern Uganda the YF outbreak occurred seven years prior to the study and some participants may not accurately recall the events. All the interviews and discussions were nonrecurrent and lasted between 30 and 60 minutes with an average of 43 minutes. Interviews and FGDs were audio-recorded, translated, and transcribed word for word. Data collection continued until theoretical saturation was reached, meaning that no new categories or insights would emerge from continued interviews, data analysis, and interpretation [38]. All transcripts were checked for completeness before qualitative content analysis with Atlas.ti software.

## Data analysis

We used a grounded theory approach inspired by Strauss & Corbin for data analysis [39]. Data collection and analysis were carried out simultaneously while still in the field, allowing us to incorporate emerging themes and refine the interview questions. In subsequent interviews and discussions, new findings were tested to confirm or refute the researchers' understanding. The results of the interviews and FGDs were discussed with the field assistants to reduce misunderstandings. The first author (LH) collected the data, coded, analyzed, and drafted the manuscript. Field assistants assisted in the process. LH conducted the coding and data analysis in English with the guidance of the supervisor, RK. The data were managed with Atlas.ti version 8.4.3 software [40]. Inductive coding progresses through open, axial, and selective coding

[30]. Through constant comparisons between data sources, similarities emerge, categories and core concepts develop, and ultimately, a theory is generated [41]. Analytical memos were used to keep track of new ideas and findings. We used multiple data collection tools to ensure that the findings could be triangulated and that their reliability could be tested [42]. The findings were shared and discussed in depth with the Ugandan research team. The interview quotes were slightly edited to improve readability.

## Results

Five main categories emerged from the analysis: perceptions of yellow fever, perceived transmission pathways, clinical presentation, challenges in obtaining a diagnosis, and treatment-seeking behaviors. The participant quotations support our findings.

### Characteristics of participants

A total of 76 individuals participated in this study (Table 1). Twenty-nine participants were recruited from northern Uganda, where Uganda's largest YF outbreak occurred in 2010. Forty-seven participants (27 from central Uganda and 20 from southwestern Uganda) were enrolled in the study from regions that experienced smaller YF outbreaks in 2016. In northern Uganda, participants belonged predominantly to Nilotic language groups; in central and southwestern Uganda, they belonged to Bantu groups. The study comprised 63% female participants. A total of 20 women reported being pregnant at the time of YF vaccination in 2016. The majority of health experts (90%) were male and had more than five years of work experience. The group of experts included members of the district health offices and medical personnel.

### Perceptions of Yellow Fever

**Perceptions of disease severity.** The participants explained that this was their first YF outbreak and reactive vaccination campaign. The information reaching remote communities through awareness campaigns portrayed it as a deadly disease yet failed to provide sufficient information according to some participants. The participants reported that through awareness campaigns, they were informed of the signs of YF, that people had died in their district, that they should keep their home environment clean, and that they would die if they did not get vaccinated. As a result, the participants perceived YF as a fatal disease. A retired headmaster affected by the 2016 outbreak in southwestern Uganda elaborated:

*I don't know much about YF because it is not common in this area. What I know is that YF attacks a person like any other febrile fever, but then if the person doesn't get treatment or does not get vaccinated, that person must die. – 72-year-old male headmaster, Rukungiri*

It was widely assumed that YF could affect anyone, but several participants believed that it was more common in pregnant women and young children. A pregnant woman expressed the following:

*YF disturbs mostly pregnant women and young children. -23-year-old farmer, vaccinated during pregnancy, Rukungiri*

Another woman elaborated further:

*There are more cases in women and babies, but everybody can easily get it. When you are pregnant, your immunity is always down, so if you are bitten by a mosquito or any way you can get that disease, you can easily pass it on to the baby. So that is why it is more common in mothers and babies. – 84-year-old female farmer, Kalangala*

In addition, some participants perceived YF to be more severe in young children and that they were more likely to die. During the semi-structured interviews, the participants frequently expressed the opinion that family and community members

**Table 1. Socio-demographic characteristics of the study participants.**

| Participants of semi-structured interviews, focus group discussions and expert interviews | | | |
|---|---|---|---|
| **Semi-structured interviews (SSI n = 43)** | | | |
| **Sex:** | 33% male, 67% female | | |
| **Age:** | median age 70 years (range 21–88 years) | | |
| **Pregnant women:** | n = 13 (Masaka, Kalangala and Rukungiri district), | | |
| | median age 28 years (range 21–37) | | |
| **Marital status:** | Married: n = 33 | | |
| | Widowed: n = 7 | | |
| | Single/divorced/no data: n = 3 | | |
| **Children:** | median 6 children (range 0–30) | | |
| **Education:** | Illiterate: n = 10 | | |
| | Primary: n = 17 | | |
| | Secondary: n = 12 | | |
| | College degree: n = 4 | | |
| **Occupation:** | Farmer: n = 25 | | |
| | Retired: n = 6 | | |
| | Housewife: n = 5 | | |
| | Business: n = 3 | | |
| | Teacher: n = 2 | | |
| | Other: n = 2 | | |
| **Study site 1** | **Northern Uganda (SSI n = 15)** | | |
| **Pader District** Subcounty/ *villages:* | Latanya, Pader Town Council, Ogom; *Dego Iwayo, Olokilee Ward, Nenibaro dyek, Okomo;* | | |
| **Kitgum District** Subcounty/ *villages:* | Labongo Amida, Omiya Anyima, Lagoro; *Okidi Central, Wang Lango, Obwore East, Aloto Central;* | | |
| **Lamwo District** Subcounty/ *villages*: | Padibe Town Council, Padibe West, Paloga;*Padibe Town Council, Tol-Polo, Katebo, Marakach B;* | | |
| **Study site 2** | **Central Uganda (SSI n = 18)** | | |
| **Masaka District:** Subcounty/ *villages:* | Kyanamukaaka, Kyesiga, Masaka municipality (the boundaries referred to are prior to the city status), Buwunga, Mukungwe; *Lukode, Mulundi, Bugere, Kamugombwa-Namaseneene, Kaloddo, Bugabira;* | | |
| **Kalangala District** Subcounty/ *villages:* | Bujjumba, Mugoye; *Buggala, Kanyogoga, Bwendero, Kyagalanyi;* | | |
| | **Southwestern Uganda (SSI n = 10)** | | |
| **Rukungiri District-** Subcounty/ *villages:* | Kebisoni, Nyakagyeme, Bwambara, Bugangari, Nyarushanje, Buyanja, Buhunga *Nyamambo A, Nyamambo B, Kebisoni Town Council, Kagorogoro, Kamunyu, Nyabubare, Runyamunyu I, Shumba, Kibunda, Ryamityari;* | | |
| **Focus group discussion (n = 23)** | | | |
| **Participants** | | **Age (median)** | **District** |
| **Older men** | n = 5 | 70 years (range 62–85) | Kitgum |
| **Older women** | n = 11 | 62 years (range 60–69) | Lamwo |
| | | 54 years (range 50–65) | Masaka |
| **Pregnant women** | n = 7 | 30.5 years (range 25–34) | Rukungiri |
| **Expert interviews (n = 10)** | | | |
| **Sex:** | 90% male | | |
| **Occupation:** | DHO, Assistant DHO, Surveillance focal person, Environmental Health Officer, Healthworker, Rapid response team, Medical Officer | | |
| **Years of experience:** | < 5 years: n = 2 | | |
| | 6-15 years: n = 5 | | |
| | >15 years: n = 3 | | |

had fallen victim to or had perished of a condition that was perceived as YF. This belief was indicated by 28 out of 43 participants interviewed. One interviewed family reported that three of their members died of YF during the 2016 outbreak.

**Perceptions of disease etiology.** Several participants reported that YF was an unknown disease to them and their communities before reactive awareness campaigns were carried out in their areas. When asked why it was called YF, participants said it was named because of the "yellow changes in the body". In Luganda and Rukiga, YF is called "Omusujja gw'enkaka" and "Omushwija gw'enkaka" respectively, and in Acholi, it is called "Gengo two", meaning fever because of yellow. Several participants said they first heard about YF from healthcare workers (HCWs) and medical experts, who came up with the name YF. A medical expert explained that the clinical presentation of jaundice and other fevers was often named YF, regardless of the underlying disease.

*Others were taking it as witchcraft, but one other thing is that most of the communities here know that any jaundice is YF. If you go and interview local people, even any fever, they say, "Ah, my child has yellow fever". – male medical expert with 17 years of work experience, Pader*

The term YF is often used to describe syndromes that may or may not overlap with the biomedical definition of YF. As a result, any ailment that causes jaundice may be seen as YF. Older women from an affected community stated that the disease was previously unknown to the community and given that only one family was affected, it was initially thought that supernatural forces caused the symptoms of jaundice.

*Those victims were kept at home for a long time, as their relatives thought they were being bewitched. The victims were taken to churches to heal, but all was in vain. (...) These people died with similar signs and symptoms within a short period, and they were from the same family. That's why people thought it was witchcraft or that the family had been bewitched. - FGD with older women, Masaka*

A medical expert elaborated further:

*Because it was a new disease to them, it affected only those who worked in the pineapple gardens. So, they thought that those people were bewitched. It took us (medical experts) time to make them accept that it was a disease caused by a virus, not as they thought being bewitched. – medical expert with 7 years of work experience, Masaka*

The communities were informed that the disease was YF transmitted by mosquitoes. One family member expressed that, initially, it was challenging for them to grasp that the disease was transmitted by mosquitoes, given that the victims did not reside in the same household and that the disease had only affected their family. They were questioning the probability of such circumstances occurring.

**Perceived cause of YF outbreaks.** Individuals residing in rural communities in districts where YF outbreaks had occurred were frequently unaware of the actual cause of the outbreak. The participants expressed a variety of opinions regarding the cause of the outbreak. These included mosquito bites, contact with wild game, and poor hygiene (e.g., unboiled water and food, inadequate sanitation, and overcrowding). A man stated:

*I think that what caused the YF outbreak was congestion, many people in the same area and very poor hygiene. – 79-year-old male farmer, Kitgum*

Additionally, some participants believed that the disease had spread through the wind and people's mobility. Some people were convinced that the disease had been imported from neighboring countries through migration. In a group discussion with older men, the following was expressed:

*During the camp period, very few people had latrines. People were defecating everywhere. (...) sometimes, that area might be a habitat for flies and animals that have YF or transmit YF, and you will not know. (...) During hunger periods, you find the South Sudanese crossing into Uganda to come and settle or search for food. You will not know if they are vaccinated or have YF. - FGD with older men, Kitgum*

**Influence of health education campaigns.** As repeatedly observed among our participants, educational campaigns to inform people about YF outbreaks have often failed to reach those affected. The participants explained that health education messages focused on the signs of YF, upcoming mass vaccination, and other preventive measures. Participants relied frequently on community sources for information. Health education was spread through door-to-door dissemination in camps for internally displaced persons (IDP camps), on radio programs, and in communal places via megaphones, local leaders, and religious leaders. However, remote villagers often had no access to radio, were spread out cultivating their fields and were informed by neighbors about the ongoing YF mass vaccination, as reported by several older participants. Members from an affected community explained that their initial attribution to witchcraft became an old story once they became aware of other YF cases in different parts of the country and were informed about the upcoming mass vaccination for all residents in the district in response to an outbreak of YF. Nevertheless, particularly in remote areas, individuals did not have access to additional in-depth information on what had caused YF in their district and how it could be prevented and treated, as expressed in the FGD with older men:

*First, as Acholi, we don't know what causes yellow fever. We see the signs and people dying but don't know exactly what causes yellow fever, its treatment, and how to prevent it. - FGD with older men, Kitgum*

Overall, we observed inequality in access to health information and knowledge among the participants. Moreover, as widely noted among our participants, the dissemination of health knowledge was frequently thwarted by numerous other health campaigns. Information about YF was therefore combined with health messages from other educational programs. Consequently, participants were often unaware that many of the implemented measures were not specifically designed to prevent the transmission of YF.

## Perceived transmission pathways of YF

Our study participants identified a range of potential transmission pathways for YF, including mosquito bites, inhalation of contaminated air, close contact with an infected individual, sexual intercourse, contact with body excretions, vertical transmission from mother to child, poor hygiene practices, patients returning from health centers, and overconsumption of yellow-colored foods. Most respondents believed that YF could be caused by more than one of the aforementioned factors and that multiple modes of transmission were possible. In an FGD with older women, the following was explained:

*YF cases became high when people were confined in the camp. Overcrowding in the camp and dirty environment caused YF. Sexual intercourse causes the spreading of YF among people. It can also be gotten through breathing in the air of someone suffering from YF when you are close by. YF is a disease that you can also get from the shelter while urinating. - FGD with older women, Lamwo*

When asked in group discussion, pregnant women who were affected by the 2016 outbreak expressed their perceived transmission pathways:

*They (awareness campaigns) said you can contract it when you get close to someone with YF. Also, if you go to areas with monkeys, you can get YF when a mosquito (just the usual mosquito) bites monkeys and then bites you. If someone has YF and coughs next to you, that virus enters you through the air, and you get YF. - FGD with pregnant women, Rukungiri*

Another young woman elaborated:

*Even the old people knew that YF was there. Someone could stay with YF for all their life, treating it. Someone could be ok. When time passes, it comes back and is translated through genes. That is what I know. They are saying this one has YF because his mother also has YF. – 23-year-old teacher, vaccinated during pregnancy, Masaka*

Group discussions elucidated the perceived potential vectors of the disease. These included bites from different mosquitoes, bites from tsetse flies, and fecal food contamination by flies.

*A dirty environment can cause YF, such as not having a latrine in your home. Because the flies like feces. They will contaminate human food with it (the feces), which can cause diseases like YF. - FGD with older men, Kitgum*

*YF is spread through mosquitoes. It can be spread by kivu (kivu = Luganda name for tsetse fly). It is a fly found in cattle. We only see them when we fetch firewood in the forests, but they are not around home. - FGD with older women, Masaka*

Another older man stated:

*People think the flies transmit YF, but I think certain types of mosquitoes cause it. It's a different type of mosquito, not the type that brings malaria. So it is through its biting that YF is caused. It sucks your blood; if you have YF, it will fly away and bite someone again, thus spreading the disease. - FGD with older men, Kitgum*

## Clinical presentation of YF

The majority of participants knew the signs and symptoms of YF. The disease was described as causing jaundice and scleral icterus, which manifests as the eyes and body turning yellow. Additionally, other symptoms, such as high fever and the excretion of bodily fluids, including vomiting, stool, and urine, turning yellow, were identified. At times, weakness and body aches were also reported.

*When the temperature rises and the eyes change yellowish, the body also changes yellowish. That's when you know that it is yellow fever. You get, uhm, headache, heartburn, pains in the joints, arms, and abdominal pain, and all your vomit will be yellowish. – 68-year-old female farmer, Pader*

Overall, the scleral icterus was perceived as pathognomonic for suffering from YF. Furthermore, participants expressed a high degree of confidence in their ability to identify family members with YF based on the presence of jaundice.

*The eyes must turn yellow because that's the only way you will know it is yellow fever. – 77-year-old male retired employee, Rukungiri*

*I will see the eyes turning yellow or them (family members) passing out yellow urine, and I will know that one is yellow fever. – 23-year-old farmer, vaccinated during pregnancy, Rukungiri*

*When someone gets yellow eyes without vomiting or fevers, I will know that it is not a common fever or condition, and I will call it yellow fever. – 37-year-old teacher, vaccinated during pregnancy, Rukungiri*

However, the clinical presentation of YF was often not distinguished from other biomedical causes of jaundice, such as newborn jaundice, severe malaria, and hepatitis. One medical expert stated that, in his experience, communities were initially unaware that YF was a distinct disease and perceived any clinical form of jaundice as YF. The expert further

elaborated that YF was a novel phenomenon for them and that he believed that awareness campaigns had facilitated an understanding of its separate disease entity.

*People had not heard about YF, but they were confusing it with jaundice, where someone becomes yellowish. But later, from mass health education done through radios and people moving from door-to-door education, people came to know that this (YF) is quite different from what the traditions think. – male medical expert with 20 years of work experience, Kalangala*

In one of the FGDs, a young woman stated that awareness campaigns indicated that jaundice and scleral icterus were signs of YF. The woman then inquired about the means of distinguishing between the causes of jaundice, as she had recently heard from other community members that there is a potential association between jaundice and liver disease in children.

*They (educational campaigns) used to tell us that the signs of YF include yellow eyes and the body turning yellow. Still, these days, other people say that, for example, children with liver disease can also get yellow eyes, so how best can you identify that someone has YF? - FGD with pregnant women, Rukungiri*

## Challenges in obtaining a diagnosis

An older woman narrated that three of her family members had died from YF during the 2016 outbreak. In February, one family member developed scleral icterus, jaundice, fever, vomiting, and weakness. The patient was taken to the nearest health facility and then referred to several hospitals, including a district hospital, the district referral hospital, and ultimately, the national referral hospital in Kampala. Shortly after, in March, two other family members developed similar symptoms and were taken to local health clinics, the district referral hospital, and churches for prayers. All three family members initially presented with high fever, abdominal pain, and jaundice. They then developed symptoms consistent with viral hemorrhagic fever (VHF) and died of an undiagnosed illness within three weeks of becoming symptomatic.

Community members explained that it was a widely held belief within their community that the affected family had been bewitched. This assumption was based on the similarity of symptoms, their occurrence within a short period, the fact that the victims came from the same family, and the observation that only those who worked in pineapple gardens were affected. A family member provided an account of the initial attribution of the phenomenon to witchcraft, stating that they had no basis for someone targeting them with witchcraft, but everyone around them was convinced it must be witchcraft, and so they followed this belief.

A health expert elaborated further that four additional cases with similar symptoms were reported within the same subcounty. The initial tests for VHFs yielded negative results. Two weeks later, the results returned positive for YF; by that time, the victims had already succumbed to their illness. The health expert explained that it was an unexpected test result, given that no YF cases had been reported in the central region of Uganda for 40 years. As a result, because the clinical presentation of YF was initially not perceived as an infectious disease, seeking medical care was delayed. Additionally, the diagnosis of YF was challenging, with patients being admitted to multiple hospitals and difficulty detecting the virus in blood samples.

## Treatment-seeking behaviors

The participants provided information regarding their treatment practices for what they perceived as YF. The described treatment-seeking behaviors encompassed a range of approaches, including traditional herbal medicine with reliance on

herbal medicine alone, neoadjuvant and adjuvant herbal medicine, and treatment from health centers (as shown in Figs 1 and 2. Treatment-seeking behavior for suspected YF).

Eight participants (six older individuals from northern Uganda, one young woman from central Uganda, and one older man from southwestern Uganda) provided first-hand accounts of their experiences contracting and surviving what they presumed to be YF. Three of the participants who had been diagnosed with YF received treatment at the hospital. They reported that no blood tests were conducted, and the diagnosis was based on clinical observation of signs and symptoms. The other participants indicated that they relied on traditional herbal medicine native to their region to treat YF. A farmer disclosed that he had contracted YF during the 2010 outbreak while residing at an IDP camp.

*I suffered from YF when I was in the camp in Paloga. It starts just like malaria; you start shivering, then your eyes turn yellow. If you don't get treatment immediately, it breaks all the joints. After that, you will feel like vomiting, followed by diarrhea. If it reaches diarrhea, it will be impossible to treat, and you will die. (…) I rushed to the hospital; they gave me tablets. They never took blood. It was the clear signs. If doctors or any nurse see your eyes are yellow, the health workers would spy around as well, they would grab anybody with yellow eyes even though you feel well and take them to the health facility. – 67-year-old male farmer, Lamwo*

Another older male participant reported that he initially sought treatment for YF at the hospital. However, when his health condition did not improve rapidly, he proceeded to receive additional therapy with a traditional herbalist.

*I was kept in the hospital for some time, and they were treating me, but the symptoms didn't go away immediately. So, I went to the traditional herbalists, who boiled Aloe vera and the other local herb Enteija. (…) I kept on drinking, and I got okay. – 77-year-old male retired employee, Rukungiri*

**Influence of disease perception on treatment-seeking behavior.** The participants widely expressed that traditional medicine had long been used in their communities to treat "yellow symptoms" among individuals. Moreover, YF was

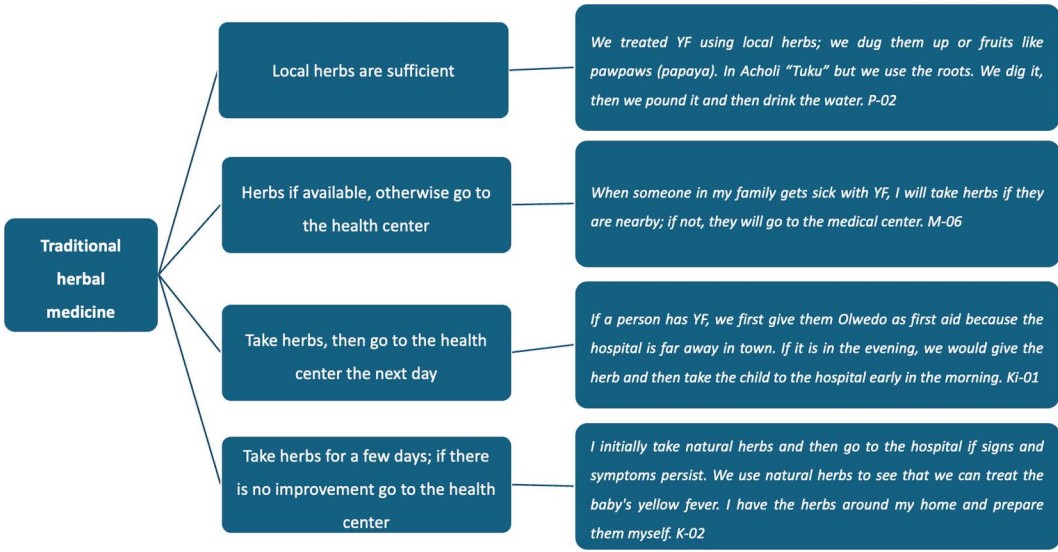

**Fig 1. Treatment-seeking behavior for suspected YF – traditional herbal medicine.**

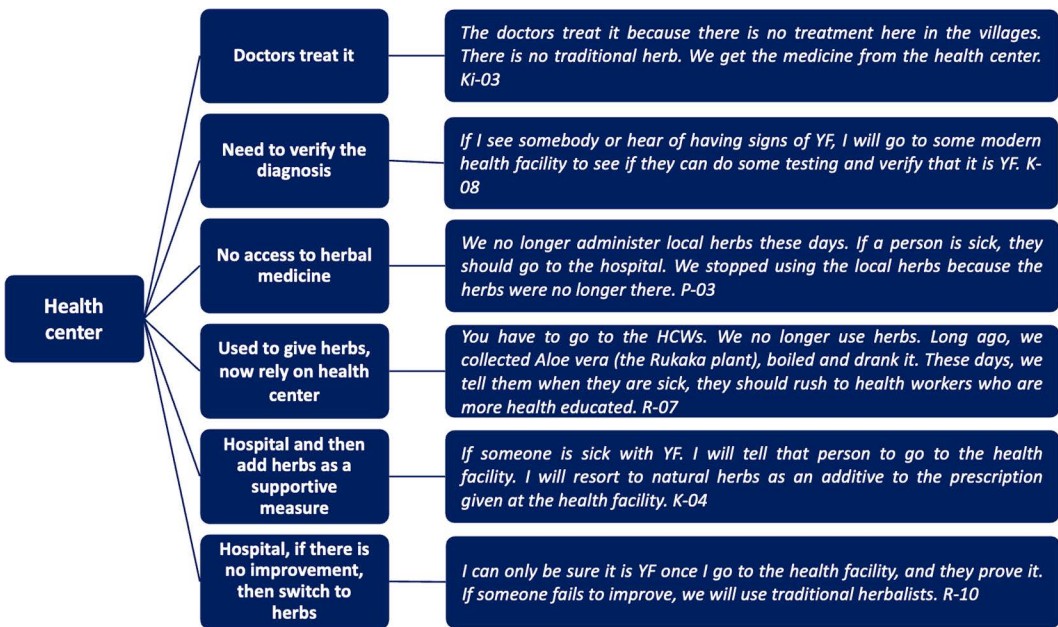

**Fig 2. Treatment-seeking behavior for suspected YF- health centers.**

frequently not distinguished from other causes of jaundice. Several participants explained that the traditional practices that use herbal remedies to treat YF are often not specific to YF. Instead, it contributes to the healing of various ailments, including jaundice, fever, malaria, headache, and abdominal complaints. As a result, it has also been used to treat the symptoms of what was perceived as YF. A pregnant woman expressed the following:

*I have heard that YF is called Omusujja gw'enkaka from the beginning. What I know is that beliefs with those traditions say that the signs are the same for them. But what they do is they get herbal medicine. – 23-year-old teacher, vaccinated during pregnancy, Masaka*

A few participants believed that a person could live with YF for a lifetime, managing it with herbal medicine rather than seeking biomedical treatment. A medical expert elaborated further:

*People confused YF with jaundice. In the beginning, older people, especially, were saying, 'No, that is, we use common herbs to treat that jaundice. This is just jaundice, which normally appears in women and so on'. – medical expert with 20 years of work experience*

The participants explained that they would use specific herbs, yellow flowers, the bark of trees, the roots of certain plants, and the leaves of green vegetables for treatment. These were mixed with water and drunk, and it was believed that this drink would flush out the "yellow symptoms" of what was perceived as YF. Older men described their experience in a group discussion:

*We had traditional herbalists who treat people with local herbs depending on the sickness. (…) They are known within the community and consulted in case of any disease outbreak. For YF, herbal medicines include, for example, "pedo, oryang, paipai". Herbs are pounded, mixed with water, and drunk to urinate all yellow from the body. - FGD with older men, Kitgum*

A recurring theme was that different ingredients were used as remedies to treat YF, with variations observed across different geographical areas. For example, in Rukungiri District, botanical substances, such as aloe vera "*Rukaka*" and a type of grass called "*Enteija*", were frequently mentioned as treatments. Overall, the participants demonstrated a comprehensive understanding of selecting, preparing, and administering herbal medicine tailored to the dosage needs of adults and children.

On the contrary, several participants indicated that although they were aware of traditional herbal medicine, they would seek medical attention from health centers if they showed signs of YF because they had been advised to use the formal healthcare system. Older women elaborated on this in a group discussion:

*Acholi knew about YF in those days. If there was YF, they had herbal medicines that they cut from the bush and gave you to drink to urinate the fever, and then you got well. But now that they have brought modern medicine to treat YF, they started saying we should not give our traditional medicine to anyone suffering from YF and that we should take them to the hospital. - FGD with older women, Lamwo*

In addition, several participants stated that they would rely on biomedical care because they believed YF is a disease that requires confirmation and treatment with tablets at the hospital. However, their understanding of YF appeared comparable to that of other participants. Overall, the participants' treatment decisions were shaped not only by how they perceived the disease, but also by availability and accessibility of healthcare facilities.

**Healthcare accessibility and trust in medical care.** As repeatedly observed among our participants, the closest health facility was often very far away, and transportation to the facility was expensive. Thus, access to health centers was limited and scarce for many of our study participants. One pregnant woman expanded upon this point, noting that, in the case of infant mortality within their communities, the cause of death is often unknown. This was due to the deceased child not being transported to a hospital for further examination and treatment, which was a consequence of limited access to medical facilities.

*You have to go to the hospital to treat YF. There are no traditional ways for YF. We term it as another condition, not YF. We treat it with local herbs. Sometimes, children die, and they will not know what killed the child because it was not taken to the hospital (…). The local herb, known as "Rukaka- that's aloe vera", we boil it, squeeze the juice, and give it to the child. (…) the child will vomit yellow stuff and then is healed. – 37-year-old teacher, vaccinated during pregnancy, Rukungiri*

Nevertheless, the majority of participants specified that they would advise family members to visit health facilities if they suspected YF. However, one woman whose family members had died of YF during the 2016 outbreak lost faith in the treatment after they died at health facilities.

*Unfortunately, it is from the health facilities that family members passed away. I don't know how it can be treated simply because the health facilities failed to see that these people could sustain their lives. – 32-year-old farmer, vaccinated during pregnancy, Masaka*

Another pregnant woman whose brother-in-law had died of YF at the hospital elaborated that smaller health facilities were insufficiently equipped to diagnose YF. When referred to a district or regional referral hospital for further evaluation, people might not be able to afford transportation, as her family experienced.

*I knew if I had signs of an illness, I could immediately visit the clinic, and they would give me drugs. Now, if I see somebody or hear of having YF, I will go instead to a modern health facility to test and verify. We need extended medical*

*services and equipped facilities (…) at clinics if you tell them your symptoms, they will only think of certain diseases and give you tablets but may not be treating the disease that is affecting you. (…) If you fall victim to YF, you are referred only to Masaka Regional Referral Hospital or Kitovu Hospital, but you realize that you never have the money to travel to those places. – 27-year-old housewife, vaccinated during pregnancy, Kalangala*

## Discussion

This study revealed that YF was often perceived as a novel and deadly disease in communities experiencing YF outbreaks. While participants generally had a good understanding of who is most susceptible to YF, their knowledge of the underlying cause of the disease, modes of transmission, and preventive strategies often overlapped with those of other diseases. This overlap was often influenced by health campaigns targeting other diseases, reflecting a vertical disease approach. Furthermore, the participants' personal experiences indicated that YF diagnosis is often lacking. In summary, those affected recognized broader correlations with YF outbreaks (e.g., higher case numbers and vulnerability due to poverty, climate change, displacement, etc.) but remained uncertain about the specifics of the disease, particularly regarding its symptoms, transmission, and prevention.

Our findings provide novel insights into how communities affected by YF outbreaks perceive and respond to these outbreaks, extending beyond existing theories on health behavior and community engagement. While many frameworks emphasize structured participation and information dissemination, our study highlights how perceptions of disease, diagnostic uncertainty, and trust in healthcare systems shape community responses during outbreaks. Limited access to in-depth health information and the absence of locally tailored health education strategies can influence community perceptions and affect prevention efforts. The overlap between YF symptoms and those of other endemic diseases, combined with limited access to health facilities and diagnostic capacity, may hinder timely detection and response. These findings underscore the need for participatory strategies tailored to local needs, particularly in contexts with multiple concurrent health threats and limited infrastructure. Our study contributes to a broader understanding of community dynamics relevant to preventing and controlling disease outbreaks by emphasizing the importance of health literacy, trust, and systemic limitations.

These findings indicate that disparities in access to health information among participants exist, leading to varying levels of understanding of YF. This was especially pronounced among marginalized groups, including individuals who were illiterate and resided in remote villages [34]. Our results showed that these vulnerable populations depended on community sources for information and lacked access to more in-depth information to educate themselves further. Awareness campaigns, which were part of the outbreak response, proved to be an effective method for disseminating general information about YF. However, these campaigns often failed to reach populations in remote areas. Moreover, the campaigns succeeded in raising awareness, but the messaging did not address the specific cause of the outbreak. Additionally, the disseminated information about YF became conflated with other health education initiatives. As a result, the participants implemented prophylactic measures that were ineffective in preventing the transmission of YF. However, it is of the utmost importance to gain an understanding of the cause of the outbreak to implement effective measures.

Our results highlighted important aspects of participants' perceptions of vulnerability to YF, particularly in communities with limited access to detailed health information. The participants identified pregnant women and young children as the most vulnerable populations for YF. This observation reflects a general understanding of health risks, which are often shaped by community knowledge and shared experiences rather than formal health education [43]. Moreover, the perception that contracting YF is linked to poverty indicators, such as inadequate sanitation, poor hygiene, overcrowded living spaces (e.g., informal settlements and congested IDP camps), and an influx of refugees from neighboring countries underscores the role of socio-economic factors in health. This suggests a broader understanding of how systemic issues contribute to disease susceptibility [44].

The literature has emphasized Uganda's high susceptibility to epidemics [45]. In addition, Uganda hosts approximately 1.6 million refugees, primarily in economically underdeveloped northern regions [18]. Furthermore, neighboring countries, such as the Democratic Republic of the Congo and South Sudan, report high incidences of YF [46], with vaccination coverage below the threshold needed to prevent outbreaks [46]. This situation heightens the risk of YF outbreaks in Uganda, particularly through the migration of unvaccinated individuals across borders [47,48].

Our findings showed that participants also linked the risk of YF to contact with wild game and forest access for occupational purposes. This indicates an awareness that livelihood activities can expose individuals to health threats [49]. The findings further highlighted profound distrust in the healthcare sector, stemming from beliefs that YF infection was attributed to a lack of hygiene practices (e.g., unsterile syringes), family members dying in health centers, and disease transmission from individuals returning from health centers. This distrust may be based on past experience [50,51].

Furthermore, the findings revealed that the clinical presentation of jaundice associated with YF was frequently not differentiated from other biomedical causes of jaundice (e.g., newborn jaundice, severe malaria, and hepatitis). Consequently, any jaundice was considered and treated as if it were YF. Moreover, the fact that other ailments can cause jaundice was often not known. However, it is crucial to consider the various differential diagnoses of jaundice to ensure an accurate diagnosis and appropriate treatment of the underlying disease. For example, just before the 2010 YF outbreak, northern Uganda experienced the world's largest hepatitis E outbreak from 2007-2009, resulting in over 10,196 cases of acute jaundice and 160 associated deaths [33]. In addition to YF outbreaks [19,21,23], Uganda has experienced numerous other VHF outbreaks in recent years, including EVD [32,52–54], Marburg virus disease [52,55–57], Crimean-Congo hemorrhagic fever [55,58,59], and Rift Valley fever [55,60–62]. Due to the similar clinical presentations of these conditions, distinguishing between them can be challenging, making laboratory diagnosis essential [63]. Thus, recognizing YF as a distinct disease entity is critical for implementing appropriate diagnostic and therapeutic measures, as well as for developing effective preventive strategies.

Our results showed that traditional herbal medicine is widely acknowledged and frequently used to treat individuals suffering from YF. This was particularly evident when the distance to the nearest health facility was far, and transportation was expensive. Research has shown that vulnerable and rural populations often have limited access to formal healthcare systems [64]. Furthermore, decisions to seek medical care for febrile illnesses in Ugandan communities are strongly influenced by cultural background, perceptions of illness severity and origin, and the healthcare system's ability to accurately diagnose and treat the illness [49].

However, diagnosing symptomatic individuals among our participants was reportedly challenging because of factors such as distance to health centers, limited blood testing, and long turnaround times. Even when medical care was accessible, a diagnosis of what was perceived to be YF was often solely based on clinical signs, such as jaundice, without confirmatory blood tests. This approach contradicts WHO guidelines, which recommend laboratory testing to confirm suspected YF cases [65]. In addition, diagnosing YF is crucial for surveillance, outbreak detection, and control efforts [3]. However, at the individual level, implementing YF diagnostics may also influence health-seeking behaviors [49].

In Uganda, YF is monitored through a passive surveillance system based on reports from healthcare providers, predominantly at primary healthcare clinics [66]. In addition, YF trends are tracked in seven high-risk locations in central and western Uganda as part of a sentinel surveillance program [67]. However, previous research has shown that HCWs in Uganda often have limited knowledge about common zoonotic diseases, including YF [68,69]. Specifically, HCWs exhibited low awareness of the causative agents and diagnostic procedures for the region's most prevalent zoonotic diseases [68]. This knowledge gap impacts YF surveillance, as effective disease monitoring and zoonosis control depend heavily on frontline HCWs' understanding of these diseases [66]. An evaluation of Uganda's YF surveillance program from 2017-2022 revealed delays in case confirmation and incomplete data, comprising the program's overall efficacy and efficiency [67].

Our findings indicated that the participating medical experts lacked practical expertise in YF, relying primarily on information from medical textbooks, with minimal clinical experience in diagnosing and managing YF cases due to the

absence of documented cases for decades in the region [70–72]. During the 2010 YF outbreak, Uganda lacked in-country testing capacity, requiring specimens to be shipped abroad for confirmation, which delayed detection by 40 days [19]. Wamala et al. noted that this delay hindered timely response efforts [19]. Furthermore, during the 2016 outbreak, the index cases presenting with suspected YF were referred to multiple health centers and hospitals for further evaluation. In recent years, a YF testing capacity has been established at the Uganda Virus Research Institute. However, this information may not have reached rural health centers. As part of a nationwide network of specimen-referral systems, samples could have been dispatched to the Uganda Virus Research Institute for analysis [21]. The failure to obtain samples at the initial health facility may be attributed to a lack of training, infrastructure such as cooling, and secure sample transportation. Many smaller health centers may lack the capacity to provide these resources [73]. In addition, the turnaround time for the samples was two weeks, and by the time the blood test was positive, the patients had already succumbed to YF. The findings illustrate the challenges inherent in reaching a YF diagnosis at that time. However, due to better surveillance, the diagnosis of YF may be more rapid in regions where VHFs such as EVD and Marburg are prevalent.

Furthermore, the absence of a rapid diagnostic process and the fact that no specific treatment for YF is currently available may result in the detection and admission of only severe cases to health centers. Given that severe cases of YF have a high mortality rate [6], this could result in the death of family members at health facilities, which has the potential to erode trust in medical services even further [74]. Importantly, in the absence of an accurate diagnosis of YF, any clinical manifestation of jaundice may be erroneously attributed to YF. This could result in the assumption that the individual had contracted YF when, in fact, another underlying cause was responsible for the jaundice. As a result, the disease may not be treated correctly. Furthermore, there is a possibility that individuals may incorrectly assume that they are immune to YF and, as a result, refrain from participating in YF vaccination campaigns in Uganda. This could impede the objectives of the EYE strategy [3].

Thus, strengthening health literacy among individuals at risk of YF is imperative to reduce health outcome inequity [75]. This could be achieved by improving health education and training, with a particular focus on vulnerable communities and frontline HCWs [75]. Health education programs should be tailored to the specific needs of the target population, given that numerous other infectious diseases with similar clinical presentations to YF are prevalent in the region. Furthermore, a participatory approach could be an essential strategy for enhancing awareness among communities at risk [76]. This approach could involve actively engaging community members in identifying their needs and preferences, and co-creating the design, development, and implementation of health education campaigns. Such collaboration ensures that health messages are culturally relevant and address local concerns. Partnering with local leaders and trusted community HCWs could further enhance trust and acceptance. Furthermore, incorporating feedback loops from community members could also help refine and adapt educational materials, thereby improving program effectiveness over time. This level of involvement promotes community ownership of the health program, builds trust, and contributes to more sustainable and impactful health education campaigns [77,78].

In Uganda, the prevailing public health model is oriented towards surveillance and case management, with a notable absence of emphasis on community engagement as a core component of a comprehensive model [49]. However, an alternative approach would be to implement horizontal programs that integrate the delivery of care for a range of ailments across different sites, thereby pooling resources and infrastructure [79]. For example, during the EVD outbreak in Uganda, a screening program for EVD was introduced at HIV outpatient clinics [80].

## Strengths and limitations

The uniqueness of this manuscript is our exploration of the perceptions of vulnerable groups in rural and remote villages regarding disease perception and treatment practices for YF. This study provides unique and novel insight into how people's perspectives are affected by YF outbreaks. We used methodological triangulation to increase the robustness of our findings through data collection from semi-structured interviews, expert interviews and FGDs. A particular strength is that semi-structured interviews and FGDs were conducted in native languages (Luganda, Rukiga, and Acholi) by field assistants familiar with the study sites. This allowed us to explore participants' disease perceptions and treatment-seeking

behavior in a culturally sensitive way. Nevertheless, our study has several limitations. The most significant limitation is recall bias. We must consider that we interviewed older people with less-than-perfect memories. Notably, at study site 1, we interviewed participants seven years after the YF outbreak. Thus, they might not have recalled events in detail, compromising the data's reliability. Second, as this study was part of a broader investigation of YF in Uganda, only individuals who had been vaccinated against YF during emergency mass vaccination campaigns as part of reactive immunization initiatives in response to outbreaks in 2010 and 2016 were included. Consequently, we may have included participants in the study who had a more comprehensive understanding of the disease and its treatment practices as they had been exposed to YF outbreaks. Third, we have to consider the time gap between data collection and publication of the results. Since then, numerous YF outbreaks have been reported in Uganda and awareness as well as perceptions of the disease may have increased and changed over time. Thus, our results reflect only the perceptions of the people who participated in this study. Furthermore, gender dynamics within society may have influenced data collection, as male field assistants aided in conducting interviews and FGDs with pregnant women. There may have been hesitancy in disclosing specific perspectives. In addition, power dynamics through an outsider, a white female researcher, may have shaped the understanding of the research findings. This study included rural and remote villagers, who were often illiterate and without access to electricity. Given these challenging circumstances, we did not follow up with them to conduct member-checking.

## Conclusion

Limited access to in-depth health information and diagnostic resources may influence community perspectives on YF outbreaks, potentially delaying and reducing the effectiveness of public health responses. Enhancing health literacy through locally tailored health education and participatory approaches by actively involving community members and trusted HCWs in at-risk communities may enable more informed prevention and response efforts. Given the similarity of YF symptoms to those of other endemic diseases, strengthening clinical training and diagnostic capacity in affected areas is essential to support accurate diagnosis and timely response to a disease outbreak.

## Acknowledgments

We want to thank the District Health Officers and Local Chairpersons of the Districts Masaka, Kalangala, Rukungiri, Kitgum, Lamwo, and Pader for granting access to their districts and communities. We would also like to thank all other members of the District Health Teams and Village Health Teams for their assistance during data collection. Furthermore, we thank Aban Kihumuro and Denis Lukwago for their support.

## Author contributions

**Conceptualization:** Lena Huebl, Aloysious Nnyombi, Eddy Walakira, Ruth Kutalek.

**Data curation:** Lena Huebl.

**Formal analysis:** Lena Huebl, Ruth Kutalek.

**Methodology:** Lena Huebl, Patricia Apoko, Denis Okello, Ruth Kutalek.

**Project administration:** Lena Huebl.

**Software:** Lena Huebl.

**Supervision:** Eddy Walakira, Ruth Kutalek.

**Writing – original draft:** Lena Huebl.

**Writing – review & editing:** Lena Huebl, Aloysious Nnyombi, Patricia Apoko, Denis Okello, Eddy Walakira, Ruth Kutalek.

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
