## [Decision Letter · Decision Letter 0]

Community Perceptions of Yellow Fever and its Treatment Practices in Regions with Reported Outbreaks in Uganda: A Qualitative Study

Dear Dr. Huebl,

Thank you for submitting your manuscript to PLOS Neglected Tropical Diseases. After careful consideration, we feel that it has merit but does not fully meet PLOS Neglected Tropical Diseases's publication criteria as it currently stands. Therefore, we invite you to submit a revised version of the manuscript that addresses the points raised during the review process.

Please submit your revised manuscript within 60 days May 25 2025 11:59PM. If you will need more time than this to complete your revisions, please reply to this message or contact the journal office at plosntds@plos.org. Please include the following items when submitting your revised manuscript:

We look forward to receiving your revised manuscript.

Kind regards,

Jin-xin Zheng

Academic Editor

Michael Holbrook

Section Editor

Shaden Kamhawi

co-Editor-in-Chief

Paul Brindley

co-Editor-in-Chief

**Reviewers' Comments:**

Reviewer's Responses to Questions

**Key Review Criteria Required for Acceptance?**

**Methods:**

-Are the objectives of the study clearly articulated with a clear testable hypothesis stated?

-Is the study design appropriate to address the stated objectives?

-Is the population clearly described and appropriate for the hypothesis being tested?

-Is the sample size sufficient to ensure adequate power to address the hypothesis being tested?

-Were correct statistical analysis used to support conclusions?

-Are there concerns about ethical or regulatory requirements being met?

Reviewer #1: 1. **Methodological Clarity**

- Specify the average interview duration

- Provide evidence of interview guide validation

- Clarify sampling strategy (purposive sampling, snowball sampling)

- Explain the method for determining interview saturation

2. **Research Rigor**

- Conduct researcher triangulation to enhance methodological reliability

- Clearly delineate novel insights beyond existing theoretical frameworks

Reviewer #2: The objectives are clear. The methodology does not entail testing a hypothesis and as such this is not applicable. The study design is appropriate and population well defined. Sample size calculations are also not necessary - point of theoretical saturation was used (though this is not referenced). No ethical concerns.

**Results:**

-Does the analysis presented match the analysis plan?

-Are the results clearly and completely presented?

-Are the figures (Tables, Images) of sufficient quality for clarity?

Reviewer #1: 1. **Presentation and References**

- Review and justify the necessity of each figure

- You don't seem to cite your previous qualitative research on vaccines

2. **Analytical Depth**

- Narrow the research focus to provide more nuanced and in-depth analysis

- Strengthen the connection between community perceptions and actual treatment practices

Reviewer #2: The organization of the themes requires some attention - see the attached pdf

**Conclusions:**

-Are the conclusions supported by the data presented?

-Are the limitations of analysis clearly described?

-Do the authors discuss how these data can be helpful to advance our understanding of the topic under study?

-Is public health relevance addressed?

Reviewer #1: The manuscript provides insights into community perceptions and treatment practices of yellow fever in Uganda. While the study offers valuable perspectives, the research would benefit from a more focused approach, concentrating on a specific core research question to enhance readability and impact.

Reviewer #2: The conclusions are relatively sound though I suggest that they could extended somewhat to discuss how they can be applied.

**Editorial and Data Presentation Modifications?**

Reviewer #1: (No Response)

Reviewer #2: (No Response)

**Summary and General Comments:**

Reviewer #1: (No Response)

Reviewer #2: A very worthwhile analysis of an under-studied area - although the implications of the gap between data collection and submission could be more clearly addressed. Some improvements could be made in the communication of the findings, through adjusting the structure of the results section - see the annotated pdf.

PLOS authors have the option to publish the peer review history of their article (what does this mean? ). If published, this will include your full peer review and any attached files.

**Do you want your identity to be public for this peer review?** For information about this choice, including consent withdrawal, please see our Privacy Policy .

Reviewer #1: No

Reviewer #2: No

**Figure resubmission:**

**Reproducibility:**



---

## [Decision Letter · Decision Letter 1]

Dear Dr. med. univ. Huebl,

We are pleased to inform you that your manuscript 'Community Perceptions of Yellow Fever and its Treatment Practices in Regions with Reported Outbreaks in Uganda: A Qualitative Study' has been provisionally accepted for publication in PLOS Neglected Tropical Diseases.

Best regards,

Jin-xin Zheng

Academic Editor

Michael Holbrook

Section Editor

Shaden Kamhawi

co-Editor-in-Chief

Paul Brindley

co-Editor-in-Chief

Reviewer's Responses to Questions

**Key Review Criteria Required for Acceptance?**

**Methods**

-Are the objectives of the study clearly articulated with a clear testable hypothesis stated?

-Is the study design appropriate to address the stated objectives?

-Is the population clearly described and appropriate for the hypothesis being tested?

-Is the sample size sufficient to ensure adequate power to address the hypothesis being tested?

-Were correct statistical analysis used to support conclusions?

-Are there concerns about ethical or regulatory requirements being met?

Reviewer #1: (No Response)

Reviewer #2: Yes/NA

**Results**

-Does the analysis presented match the analysis plan?

-Are the results clearly and completely presented?

-Are the figures (Tables, Images) of sufficient quality for clarity?

Reviewer #1: (No Response)

Reviewer #2: Yes/NA

**Conclusions**

-Are the conclusions supported by the data presented?

-Are the limitations of analysis clearly described?

-Do the authors discuss how these data can be helpful to advance our understanding of the topic under study?

-Is public health relevance addressed?

Reviewer #1: (No Response)

Reviewer #2: Yes

**Editorial and Data Presentation Modifications?**

Reviewer #1: (No Response)

Reviewer #2: (No Response)

**Summary and General Comments**

Reviewer #1: (No Response)

Reviewer #2: (No Response)

PLOS authors have the option to publish the peer review history of their article (what does this mean? ). If published, this will include your full peer review and any attached files.

**Do you want your identity to be public for this peer review?** For information about this choice, including consent withdrawal, please see our Privacy Policy .

Reviewer #1: No

Reviewer #2: No

---

## [Editor Report · Acceptance letter]

Dear Dr. med. univ. Huebl,

We are delighted to inform you that your manuscript, "Community Perceptions of Yellow Fever and its Treatment Practices in Regions with Reported Outbreaks in Uganda: A Qualitative Study," has been formally accepted for publication in PLOS Neglected Tropical Diseases.

Best regards,

Shaden Kamhawi

co-Editor-in-Chief

Paul Brindley

co-Editor-in-Chief
